# Fate of Saxitoxins in Lake Water: Preliminary Testing of Degradation by Microbes and Sunlight

Niels O. G. Jørgensen [1,*], Raju Podduturi [1], Charlotte Frydenlund Michelsen [2], Thea Jepsen [1] and Munique de Almeida Bispo Moraes [3]

1 Department of Plant and Environmental Sciences, University of Copenhagen, 1871 Frederiksberg, Denmark
2 4Life Solutions ApS, Fruebjergvej 3, 2100 Copenhagen, Denmark
3 Fisheries Institute—Agência Paulista de Tecnologia dos Agronegócios, Secretariat of Agriculture and Food Supply of São Paulo State, Water Resources and Fisheries Research Center, São Paulo 04014-900, Brazil
* Correspondence: nogj@plen.ku.dk

**Abstract:** The cyanobacterial toxin saxitoxin (STX) is mainly associated with the blooms of marine dinoflagellates, but it is also produced by several species of freshwater cyanobacteria. So far, the degradation of STX has only been demonstrated by physicochemical treatments, but in this study, we demonstrated that natural factors, such as bacterioplankton and sunlight, had the capacity for degrading STX in the eutrophic Lake Arresø, Denmark. Natural lake bacterioplankton could reduce STX concentrations by 41–59%. A similar reduction was shown for four saxitoxin analogs. The exposure of the lake water to natural sunlight or simulated sunlight also reduced both intracellular and extracellular, dissolved STX. During 4–8 h exposure, natural sunlight reduced intracellular STX by 38–52% but increased extracellular, dissolved STX by 7–29%. Corresponding values for simulated sunlight were a reduction in intracellular STX by 16–45% and increased levels of extracellular STX by 10–33%. In particle-free lake water, the two types of sunlight reduced ambient, dissolved STX by 13–17%. The light exposure was observed to damage >94% of the *sxtA* gene involved in STX synthesis. This study demonstrated that lake water bacterioplankton and sunlight can modify STX by degradation and cell destruction, and that the biosynthesis of STX may be inhibited by exposure to sunlight.

**Keywords:** saxitoxin; lake water; microbial degradation; UV degradation; *sxtA* gene

## 1. Introduction

The neurotoxin saxitoxin (STX) is produced by marine dinoflagellates and causes paralytic shellfish poisoning (PSP) in humans and animals [1], but some freshwater cyanobacteria have also been identified as STX producers. The first observed freshwater producer of STX was the cyanobacterium *Anabaena circinalis* [2], but later, several genera of freshwater cyanobacteria have been recognized as producers of STX or its analogs [3]. Identified STX producers in freshwater include the genera *Raphidiopsis* [4], *Dolichospermum* [2], *Microseira* [3,5] and *Cuspidothrix* [6].

Most studies on STX and STX analogs in natural waters have focused on intracellular pools of the toxins. Casali et al. [7] measured 5 to 28 fg of STX per trichome (filament) of *R. raciborskii*, while five different STXs together were found to vary from 9 to 142 fg per phytoplankton cell or colony in lakes in Denmark, with *Dolichospermum* as the dominant cyanobacterial genus [8]. When related to the volume of water instead of cellular units, intracellular concentrations of STXs were reported to range from 66 to 985 ng of STXs L$^{-1}$ (sum of STX and four analogs) in Danish lakes [8], from 5 to 7 µg L$^{-1}$ (sum of STX and the analog NEO) in a shallow reservoir in France [9] and from 33 to 1070 µg STX L$^{-1}$ during blooms in Finnish freshwater [10]. The differences observed in concentration ranges from the different freshwaters may reflect actual biological differences, such as bloom episodes, or the applied analytical procedures.

When intracellular STXs are released into water, either due to cell damage, mortality or by active transport [11], they constitute a large potential source of dissolved toxins that may exceed the recommended WHO guideline value of about 3 µg L$^{-1}$ [12]. This poses a risk on human and animal health, but until now, the knowledge on the occurrence of dissolved, extracellular STX and its seasonal variation in freshwaters is scarce. In the Brazilian Alagados freshwater reservoir, highly variable seasonal concentrations of dissolved STX and STX analogs were observed. In 2007–2008, dissolved STXs ranged from 5 ng L$^{-1}$ in spring to 44 and 51 ng L$^{-1}$ in summer and autumn, respectively [13], but significantly higher concentrations were measured in 2013–2014, ranging from 0.36 µg L$^{-1}$ in spring to 5.3 µg L$^{-1}$ in autumn [14]. Since the STX-producing *Raphidiopsis raciborskii* was the dominant cyanobacterium in the Alagados reservoir, blooms of this species most likely caused the high STX concentrations [14].

The presence of STXs in natural waters reflects a balance between the input and removal of the STXs. While the release of intracellular STXs can increase the level of dissolved, extracellular STXs, degradation processes may reduce or transform the STXs. However, as concluded in a review by Kumar et al. [15], STXs appear rather resistant to biological degradation. Supporting this, Ho et al. [16] did not detect any degradation of STX in reactors with natural freshwater from locations with an expected production of cyanotoxins. In contrast to STX, the STX analog gonyautoxin (GTX) was found to be degradable by microorganisms in a sand filter with sub-surface irrigation drainage water with a half-life of 9 to 28 days at 25 °C [17]. Some microorganisms appear also capable of transforming STXs. Kayal et al. [18] observed the changes in five STXs in biologically active sand filters, where less toxic STX analogs were converted to more toxic analogs of GTX, but a slight increase in the STX concentration also occurred.

Bacteria living in association with marine invertebrates have also shown potential for transforming STXs. Thus, bacteria isolated from marine bivalves were capable of converting various GTXs into other STX variants [19]. In natural waters, STXs may also be removed by non-biological processes, e.g., when saxitoxin-containing cells settle out on the bottom of the water column and adsorb to sediment particles where it may persist for years [20].

When STXs and other cyanotoxins occur in source water to be used as potable water, various physicochemical treatment processes, e.g., photocatalysis, ozonation and chlorination, can be applied to reduce the toxin concentration [15,21]. While high-intensity UV radiation in photolysis will destroy cyanotoxins, UV radiation at natural solar light intensity levels has been found to affect production and release of cyanotoxins in living cells. Beamud et al. [22] exposed the cyanobacterium *R. raciborskii* to short time intervals (30 min to 1 h) with UV-A and UV-B radiation at 0.3 and 0.8 W m$^{-2}$ (for comparison, a UV index of 10 in the open sky equals 0.25 W m$^{-2}$), respectively, and observed that the UV radiation changed the growth and morphology of the cyanobacterium, and it also stimulated the expression of a core gene in STX production (*sxtU*) and intracellular concentrations of STX. The mechanisms causing this increased STX production in the *R. raciborskii* cultures remain to be determined, and it is unknown if a similar UV stimulation occurs in natural environments. Yet, there is no documentation for such effects from natural solar radiation on gene expression or ambient concentrations of saxitoxins [23].

As indicated above, both biological and non-biological processes may cause degradation or transformation of STXs, but the impact of these processes in freshwaters remains to be studied. Therefore, in this study we examined whether indigenous microorganisms in a eutrophic lake had the capacity to degrade ambient concentrations of STX and some STX analogs. In addition, we examined the effects of solar radiation (both under natural and simulated sunlight conditions) on the release and degradation of STXs in the lake water by utilizing transparent, commercial SaWa® plastic "bags" (produced by 4Life Solutions ApS, Copenhagen, Denmark). Possible light-induced changes in the concentration of STX, and in some cases also the analogs dcNEO, dcSTX, GTX2/3 and NEO, were determined by either HPLC fluorescence detection or the ELISA method. These five toxins were selected because they are reported to be among the most common saxitoxin analogs and they include the

dominant toxic types (dcSTX, NEO and STX) [24]. The effect of solar radiation on the abundance of a core gene in the biosynthesis of STX, *sxtA*, was determined by qPCR and was related to intracellular concentrations of STXs.

## 2. Material and Methods

### 2.1. Sampling and Location

Water was collected on six different dates in 2020 (21 June, 7 and 28 July, 17 August, 15 September and 11 October) and three dates in 2021 (6 June, 4 July and 13 September) at 0.3–0.4 m depth in the shallow eutrophic Lake Arresø (55°58′28.1″ N, 12°07′03.6″ E; area of 3987 ha; maximum and mean depths of 5.6 and 3.1 m, respectively) in northern Zealand, Denmark. Water temperatures varied from 10.4 °C in October 2020 to 24.1 °C in August 2020. Additional information on the lake and water chemistry can be found in Podduturi et al. [8].

### 2.2. Analysis of Intracellular Pools and Dissolved, Extracellular Pools of STXs in the Lake Water

For analysis of intracellular STXs, plankton in 60 to 100 mL water was collected in triplicate on 47 mm diameter 0.2 μm cellulose nitrate filters by vacuum filtration. The filters were immediately stored folded in aluminum foils and kept frozen until extraction.

Extraction of STXs was performed by sonication of the filters in 1.5 mL 0.5 M acetic acid in 2 mL glass vials according to procedure by Podduturi et al. [8]. Briefly, a Branson SFX 250 Sonifier (Branson Ultrasonics, Brookfield, Connecticut, USA) equipped with a 3 mm micro-tip was applied for 4 min at intensity level 5 while the vials with filters were kept in an ice bath. Next, particles in the sonicated material were removed by centrifugation at $10,000 \times g$ for 5 min and, finally, the supernatants were filtered through a 0.2 μm pore size 8 mm diameter syringe filter before analysis by HPLC.

Aliquots of the filtrates from the filtration of plankton (see above) or degradation studies (see below) were collected and stored frozen in glass vials until analysis by HPLC or the ELISA method. To ensure that concentrations of STXs were above the analytical detection limit by HPLC (about 1 pg per 25 μL injection, corresponding to 40 ng $L^{-1}$) or ELISA (about 10 ng $L^{-1}$), STXs in the water samples were concentrated by freeze-drying. Typically, 10 mL water was freeze-dried, followed by addition of 10 mL of the same water sample. To ensure redissolution of the freeze-dried material in the added lake water, the samples were lightly treated in a sonicator bath for about 1 min. The freeze-drying and redissolution steps were repeated 2 to 4 times for each sample, meaning that the final samples for analysis contained material from 20 to 40 mL lake water. Finally, the freeze-dried material was redissolved in 1 mL Milli-Q water and filtered through a 0.2 μm filter cartridge before analysis by HPLC.

Saxitoxin and four of its analogs were quantified after HPLC separation and dual post-column treatment according to method of Yu et al. [25] with the modifications given in Podduturi et al. [8]. Briefly, a Waters 2695 Alliance Separation Module was applied to separate the STXs on a Waters XBridge C8 3.5 μm particle size and $4.6 \times 150$ mm column with a corresponding $3.9 \times 5$ mm guard column, and a Waters 2475 fluorescence detector (Waters Corporation, Milford, MA, USA) for quantification of the STXs. For post-column treatments, a Waters 515 pump and a Shimadzu LC-10 AD pump (Shimadzu company, Kyoto, Japan) were used. The mobile phases were: (A) 11 mM heptane sulfonate and 5.5 mM $H_3PO_4$ adjusted to pH 7.1 with $NH_4OH$; (B) 11 mM heptane sulfonate, 16.5 mM phosphoric acid and 11.5% acetonitrile adjusted to pH 7.1 with $NH_4OH$. Th epost-column reagents were: (1) 100 mM $H_3PO_4$ and 5 mM periodic acid solution, adjusted to pH 7.8 with 5 M NaOH as reagent 1; (2) 0.75 M $HNO_3$ as a solvent. Flowrates and solvent gradients were as in Podduturi et al. [8]. STXs standards were purchased from National Research Council Canada, Halifax, Nova Scotia, Canada and included decarbamoylneosaxitoxin (dcSTX), decarbamoylsaxitoxin (dcSTX), gonyautoxin- 2&3 (GTX2/3), neo-saxitoxin (neoSTX) and saxitoxin (STX).

In the degradation experiments conducted in 2021, concentrations of STX were measured by ELISA immunoassay technology using a 96-plate saxitoxins (PSP) kit from Abraxis (https://abraxis.eurofins-technologies.com (accessed 27 October 2022)). For the 3 June and 7 July samples, concentrations of STX were measured in triplicate in 0.2 μm filtered water that was concentrated 10-fold by freeze-drying and redissolved in Milli-Q water. For the September 7 samples, STX was measured in unconcentrated 0.2 μm filtered water due to the high concentrations present. According to Abraxis, cross-reactivities with STX analogs may occur. The highest interferences were (with STX being 100%): dcSTX 29%, GTX2/3 23%, GTX 23% and Lyngbyatoxin 13%. The potential effects on measured concentrations of STX from these interferences were not examined here.

### 2.3. Quantification of STX-Producing Cyanobacteria in the Lake Waters

For quantification of STX-producing cyanobacteria in Lake Arresø, plankton in 60 to 100 mL water was collected in triplicate on 47 mm diameter 0.2 μm mixed cellulose ester filters by vacuum filtration. The filters were immediately stored folded in aluminum foils and kept frozen until extraction of DNA.

DNA was extracted from the membrane using a DNeasy Power Water kit (Qiagen, Hilden, Germany) according to the manufacturer's guidelines, including the lysis method as suggested for algal samples (incubation of membrane filter with lysis buffer for 10 min at 65 °C). The quantity and quality of the extracted DNA was assessed by NanoDrop™ spectrophotometry (Thermo Fisher Scientific, Wilmington, NC, USA), where a 260/280 ratio was used to assess the quality of DNA. The quantitative PCR (qPCR) primer set targeting the *sxtA* gene in cyanobacteria by Podduturi et al. [8] (*sxtA*-RF1 (ACAAACCGGC-GACATAGATG) and *sxtA*-RR1 (TTTCCCGATCTGCCAGCTTA)) was used to determine copy numbers of *sxtA* in the lake water samples. The amplification protocol followed Podduturi et al. [8], except that an Agilent AriaMx instrument (Agilent Technologies, Santa Clara, California, USA) was applied. DNA extracted from a culture of *Aphanizomenon gracile* (NIVA-CYA 851, provided by The Norwegian Culture Collection of Algae (NORCCA), Oslo, Norway, https://niva-cca.no (accessed 28 October 2022)) was used as a calibration curve to quantify *sxtA* copies in lake water samples. Cells of *A. gracile* were counted by epifluorescence microscopy after staining with SYBR Green 1 (www.thermofisher.com (accessed on 28 October 2022)) as shown below.

### 2.4. Bacterial Degradation of STXs in the Lake Water

The capacity for bacterioplankton in Lake Arresø for degradation of STX and its analogs was tested in July, August and September 2020. Triplicate portions of 90 mL lake water filter-sterilized through 0.2 μm sterile filter cartridges was mixed with GF/C-filtered lake water (assumed pore size of 1.3 μm, i.e., only bacteria were expected to be present) and incubated in the dark on a shaking table at slow speed at 20–22 °C. At about 12 to 24 h intervals, 10 mL water was sampled from each culture flask, 0.2 μm filtered and stored at −20 °C for later analysis of concentrations of STXs. For determination of bacterial numbers in the cultures, formaldehyde was added to 5 mL water samples to a final concentration of 2% before storage at −20 °C. Bacterial densities were determined by epifluorescence microscopy of SYBR Green-1-stained cells. To samples of 1 mL water, 10 μL of a 1:100-diluted solution of SYBR Green 1 was added and followed by 10 min incubation in the dark. Next, the samples were filtered onto 25 mm diameter 0.2 μm pore size dark polycarbonate filters. The filters were mounted on microscopy slides with a thin film of immersion oil below and above the filter before being covered by a glass slip. Fluorescent cells were counted randomly on the filters until a standard variation of about <10% was reached.

### 2.5. Isolation of STX-Degrading Bacteria from Lake Water Samples

Detection of bacteria capable of degrading STX was tested by two approaches. In the first approach, bacteria in 10 mL Arresø water (defined as particles > 0.2 μm and <1.2 μm) were collected on 0.2 μm 25 mm diameter polycarbonate filters. The bacteria were removed

from the filters with a razor blade and inoculated in triplicate into 15 mL medium containing 100 mM glucose, 50 mM asparagine, mineral salts (0.21 M $MgSO_4$, 1.12 M $KH_2PO_4$ and 0.7 M KCl) and selected trace elements according to Malz et al. [26] into 100 mL Kapsenberg flasks. In some of the incubations, addition of glucose was eliminated. STX was added to each flask to a final concentration of 16 µg $L^{-1}$. The flasks were incubated on a shaker at room temperature for 3 weeks. To detect degradation of STX, samples for analysis of STX by the ELISA method were collected every third day.

In the second approach, petri dishes were prepared with 20 mL of the same SXT-containing medium as above without glucose, but with the addition of 2% agar. The petri dishes were inoculated with 100 µL bacterial suspension, corresponding to bacteria in about 10 mL lake water, prepared by filtration and removal with a razor blade as above. The petri dishes were incubated in the dark at 20 °C and were regularly inspected for growth.

### 2.6. Effect of Natural and Simulated Solar Conditions on STX Levels Using SaWa® Bags

Effects of solar radiation on dissolved, extracellular STX levels in natural and filter-sterilized lake water were tested utilizing transparent, commercially available SaWa® Bags. A SaWa® Bag is a 4 L household water treatment device developed by 4Life Solutions ApS, formerly known as SolarSack ApS, Copenhagen, Denmark (https://4lifesolutions.com/ (accessed on 13 October 2022)) and is made of a 5-layer transparent polymer frontside where transmittance levels throughout the UV-B and UV-A range on average are equal to 83% and 88%, respectively, and with a dark-blue backside, which during solar exposure increases the water temperature to above 45 °C. A SaWa® Bag utilizes the solar water disinfection (SODIS) process, harvesting natural solar UV rays and heat, to purify drinking water sources contaminated with bacteria, viruses and protozoa and provide safe drinking water to low-income communities, where there is no or only limited infrastructure for safe drinking water access. A SaWa® Bag works where the solar irradiance levels are favorable for the SODIS process, reaching solar radiation above 500 W $m^{-2}$ in the irradiant range of 300–800 nm over a 3–5 h period, which is mainly in geographical locations between latitudes of 35° N and 35° S. Under such favorable SODIS conditions, a SaWa® Bag can purify 4 L of water in 4 h.

For the purpose of the present experiments and water sampling, miniature bags of 600 mL were designed (Figure S1). The SaWa® Bags were either filled with natural lake water to study solar radiation effects on intact phytoplankton or 0.2 µm filtered lake water to examine if the solar radiation affected concentrations of STXs in the water samples. Triplicate bags were heat-sealed before exposed to natural sunlight conditions in Denmark between 10:00 and 14:00 (6 June 2021), or between 8:00 and 18:00 (4 July and 13 September 2021), or were incubated for four hours under simulated solar irradiance conditions using a Q-sun XE-1 Xenon Test Chamber (Guangdong Sanwood Technology Co., Ltd., Dongguan, China) (www.sanwood.cc (accessed on 13 October 2022)), following the WHO recommended solar simulation settings [27], delivering an irradiant range of 300–800 nm, equivalent to 550 W $m^{-2}$, and a total UV irradiance (280–400 nm) level of 60 W $m^{-2}$.

During the outdoor exposure the temperature occasionally reached 40 °C, while a maximum of 42 °C occurred in the Xenon test chamber. In all cases, control bags placed outdoor in the shade at a temperature identical to the incubated bags, or in the dark in the laboratory as control for the simulated solar conditions, were applied.

At end of the incubations, water samples of 50 to 100 mL were collected and filtered through 47 mm diameter 0.2 µm pore size filters (for natural lake water), after which the filtrates and filters were stored at −20 °C until the analyses of dissolved, extracellular STXs and extraction of DNA from material on the filters. From the filter-sterilized water, only water samples were collected and stored frozen until analysis for dissolved, extracellular STXs.

An overview of the sampling dates, analyses and treatments performed are summarized in Table 1.

**Table 1.** Overview of sampling dates, analyses and treatments of Lake Arresø water.

| Sampling Date | Detection of SXT in Water (W) and Phytoplankton (P) | Detection of SXT Analogs in Water | Degradation of SXT by Lake Water Bacteria | Degradation of SXT by Solar Exposure | Quantification of SXT Producers by qPCR |
|---|---|---|---|---|---|
| 21 June 2020 | HPLC (W + P) | | | | Yes |
| 7 July 2020 | HPLC (W + P) | | | | Yes |
| 28 July 2020 | HPLC (W + P) | HPLC | Yes | | Yes |
| 17 August 2020 | HPLC (W + P) | | Yes | | Yes |
| 15 September 2020 | HPLC (W + P) | HPLC | Yes | | Yes |
| 11 October 2020 | HPLC (W + P) | HPLC | | | Yes |
| 6 June 2021 | ELISA (W) | | | Yes | Yes |
| 4 July 2021 | ELISA (W) | | | Yes | Yes |
| 13 September 2021 | ELISA (W) | | | Yes | Yes |

## 3. Results

### 3.1. Occurrence of STX in the Lake Water and Number of sxtA Copies

The concentrations of STX in the lake water varied between 16.3 and 24.0 ng L$^{-1}$ in six of the nine samplings in June, July and October in 2020 and 2021, but slightly larger concentrations occurred in August (41.4 ng L$^{-1}$) and September 2020 (45.5 ng L$^{-1}$) (Figure 1A). A substantially higher concentration of 442 ng L$^{-1}$ was measured on 13 September 2021.

The intracellular STX pools in particles >0.2 μm were the lowest in June and July (0.8 to 1.8 ng L$^{-1}$) in both years (Figure 1B). Higher pools of 107 to 273 ng L$^{-1}$ occurred in late summer and fall in 2020, while a maximum of 345 ng L$^{-1}$ was measured on 13 September 2021. The changes in dissolved, extracellular STX and intracellular STX at the sampling dates covaried (Pearson coefficient of 0.758; $p = 0.018$).

The number of cells in the lake water producing STX and STX analogs were determined by a qPCR assay targeting the *sxtA* gene (involved in the synthesis of all STX analogs). The *sxtA* copy number increased during summer in 2020 and 2021 but declined in October (Figure 1C). In 2020, the copy numbers ranged from $7.2 \times 10^3$ mL$^{-1}$ on 21 June to $5.61 \times 10^6$ mL$^{-1}$ on 15 September, before declining to $1.62 \times 10^6$ mL$^{-1}$ on 11 October. In 2021, the numbers increased from $0.6 \times 10^3$ mL$^{-1}$ on 6 June to $800 \times 10^3$ mL$^{-1}$ on 13 September.

To determine the intracellular content of STXs per cell, the concentration of all five STXs (GTX, NEO, dcNEO, dcSTX and STX) was related to the *sxtA* copy numbers, since *sxtA* is engaged in the synthesis of all STX analogs, as mentioned above. In 2020, the copy-specific content varied between 0.03 and 0.13 fg per *sxtA* copy on five of the six sampling dates, but a higher value of 2.92 fg per *sxtA* copy occurred on 7 July (Figure 1D). In 2021, the highest measured value of 35.5 fg per *sxtA* copy was determined on 6 June, while values of 0.47 and 0.63 fg per *sxtA* copy occurred on the remaining two sampling days. The copy-specific intracellular content did not correlate with the copy numbers, intracellular or dissolved STXs in the water (Pearson correlations of $-0.134$ to $-0.282$; $p = 0.464$ to 0.731).

### 3.2. Concentrations of STX Analogs in Lake Water in 2020

Among the studied STX analogs (GTX, NEO, dcNEO and dcSTX), the highest concentrations were found for NEO and dcNEO. NEO reached 243 ng L$^{-1}$ on 15 September, while dcNEO was highest on 17 August with 198 ng L$^{-1}$ (Figure 2). The total concentrations of the five STXs varied from 41.1 ng L$^{-1}$ on 28 July to 367 ng L$^{-1}$ on 15 September. Among the five STX analogs, the proportion of STX varied from 2.7% on 28 July to 23.9% on 11 October.

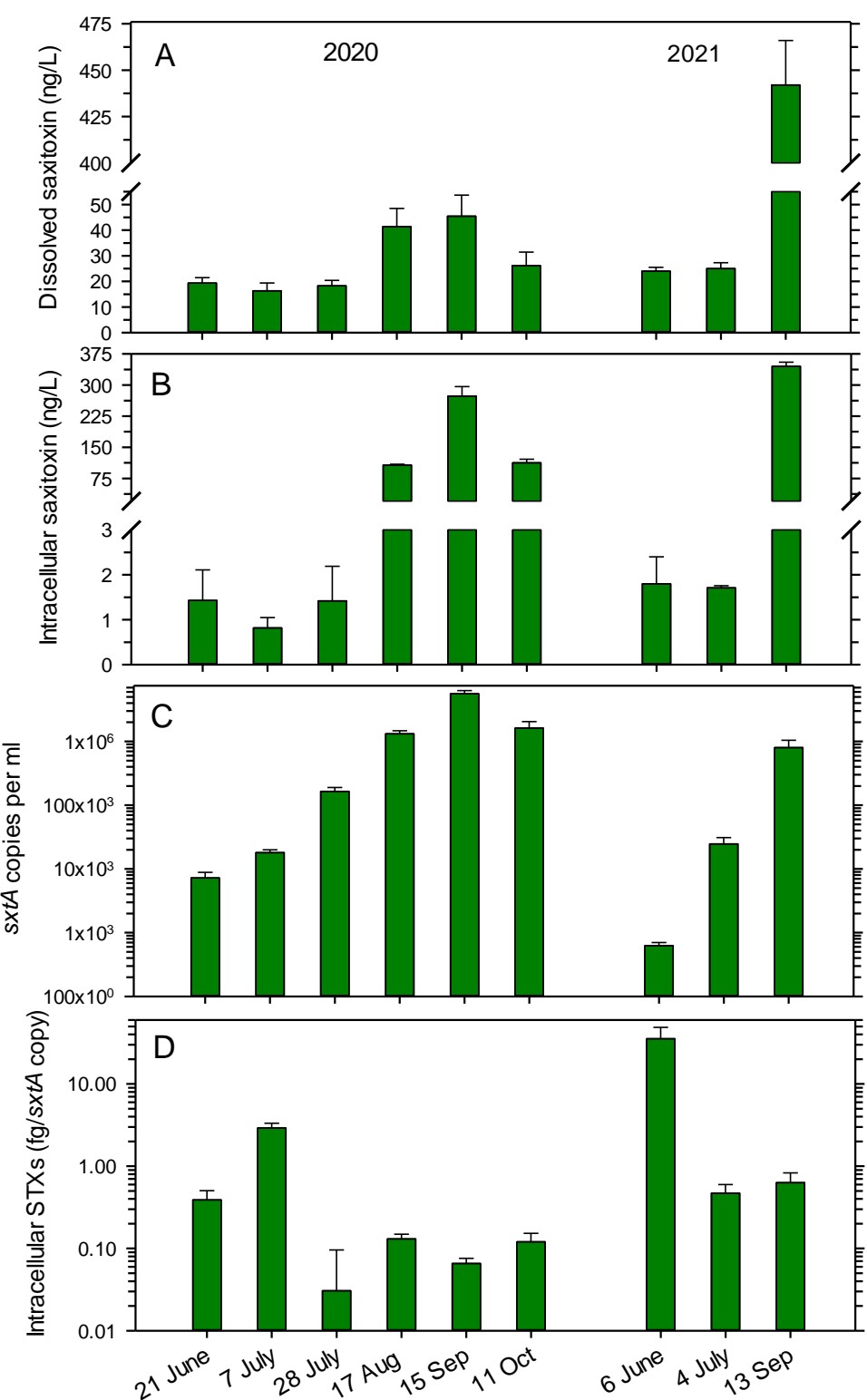

**Figure 1.** Distribution of saxitoxin (STX) in Lake Arresø in 2020 and 2021. (**A**) Concentrations of dissolved, extracellular STX in lake water; (**B**) concentrations of intracellular STX; (**C**) copy number of the *sxtA* gene; (**D**) concentration of intracellular STX per *sxtA* copy number. STX was quantified by HPLC detection. Means ± 1 STD are shown (*n* = 3).

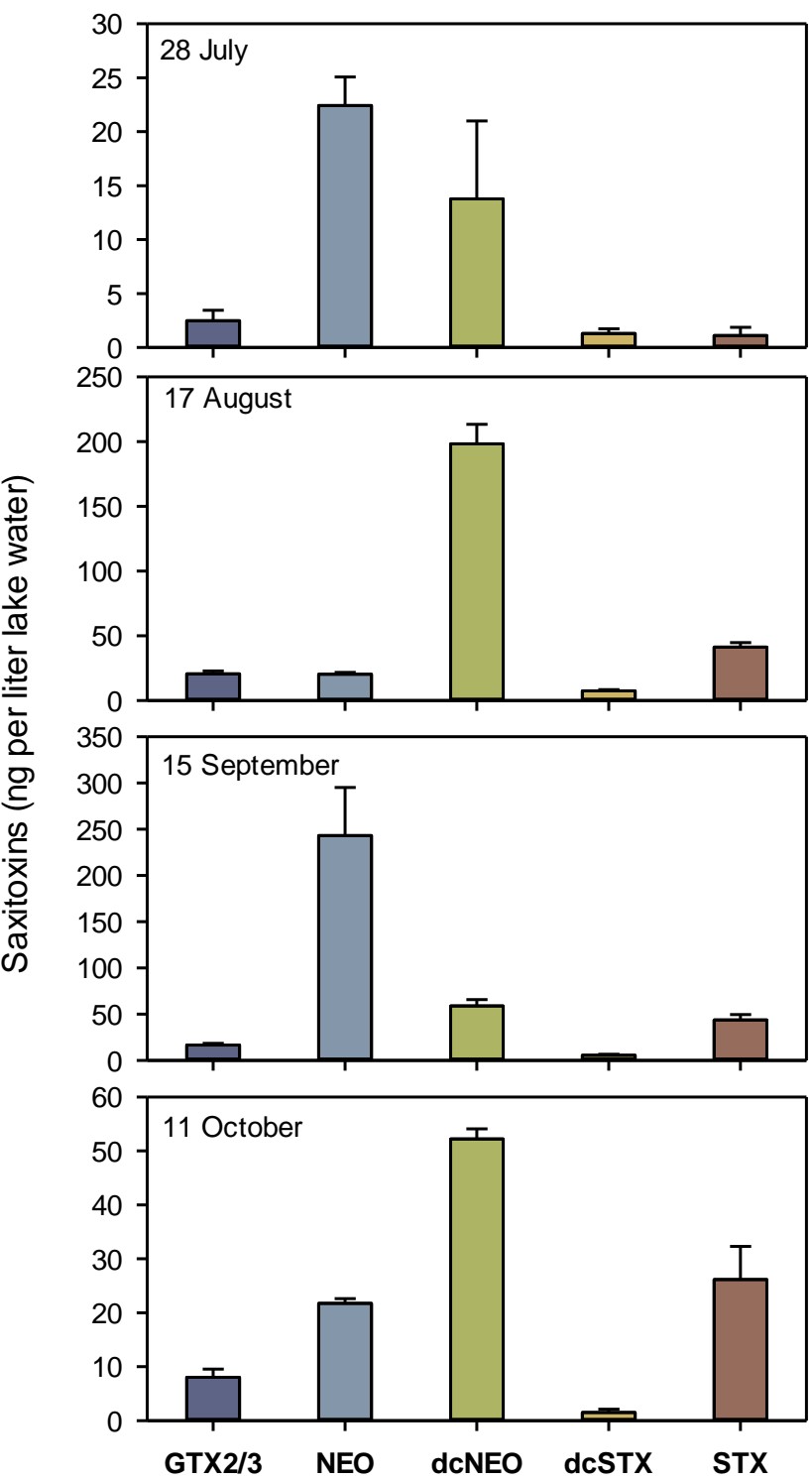

**Figure 2.** Concentrations of dissolved saxitoxin analogs in Lake Arresø in 2020, quantified by HPLC. Means ± 1 STD are shown (*n* = 3).

### 3.3. Bacterial Degradation of STX in the Lake Water

The potential for degradation of STX by lake water bacteria was tested on three occasions. The initial filtrations for the set-up of the incubations were found to not affect the STX concentrations, since STX concentrations in the original lake water and at start of the incubations were similar (Figure 3). Since the bacterial density in the incubations

slowly began declining after one day (Figure S1), the degradation experiments were not continued for longer than about 3 days.

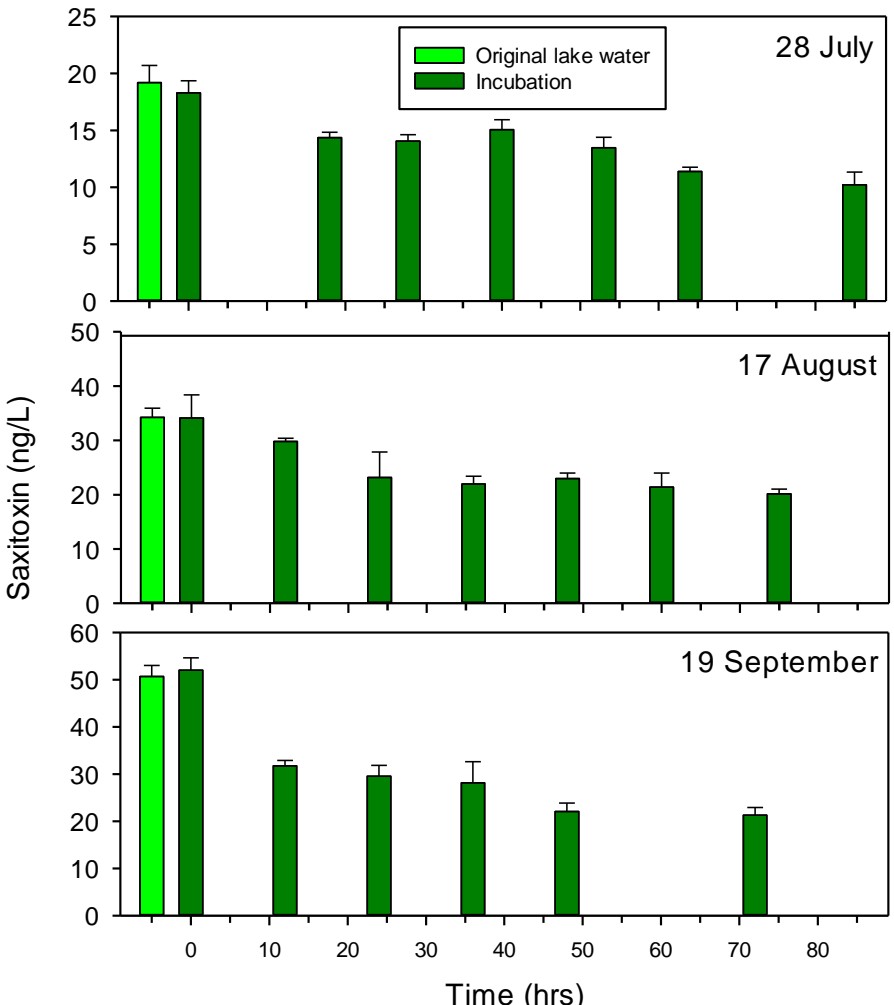

**Figure 3.** Microbial degradation of saxitoxin in laboratory cultures with Lake Arresø water in 2020. The cultures consisted of 90% 0.2 μm filtered lake water mixed with 10% lake water with particles < 1.3 μm (assumed to include bacteria only). Quantification performed by HPLC. Means ± 1 STD are shown (*n* = 3).

The incubations demonstrated a slow and partial degradation of STX, reducing STX in the water by 44.1% (from 18.3 to 10.2 ng $L^{-1}$) after 85 h on 28 July and by 40.9% (from 34.1 to 20.1 ng $L^{-1}$) after 75 h on 17 August (Figure 3). On 15 September, STX was reduced by 38.9% after 12 h, after which the decline slowly continued to a final reduction of 59.1% at 72 h, corresponding to a total reduction from 52.0 to 21.3 ng $L^{-1}$. Bacterial growth was observed in all three sets of incubations and increased the bacterial density from an initial concentration of about $2 \times 10^6$ cells per mL to about $6 \times 10^6$ cells per ml after one or two days, as exemplified for the incubations on 28 July (Figure S1). STX concentrations in the bacteria-free (0.2 μm filtered) lake water were not analyzed in parallel with sampling of the incubations with lake water bacteria. However, STX concentrations in 0.2 μm filtered lake water remained unchanged for 4 to 6 days at room temperature in the dark, suggesting that no non-biological changes in STX occurred during the up to 85 h incubation periods.

Tests for the detection and isolation of specific lake water bacteria with the capacity for degrading STX were not successful. The cultures with lake water bacteria grown in liquid mineral media with or without glucose and asparagine and enriched with 16 μg STX $L^{-1}$ did not cause any change in the STX concentration after 3 weeks. In the second approach,

the inoculation of lake water bacteria onto petri dishes with the same medium enriched with STX but without glucose, no bacterial colonies were observed after 3 weeks.

### 3.4. Sunlight-Induced Effects on STX Concentrations

The exposure of solar radiation to the lake water showed that dissolved STX in natural lake water increased by 10.4% (from 24.0 to 26.5 ng $L^{-1}$) after 4 h of exposure on 6 June and 29.2% (from 24.1 to 31.0 ng $L^{-1}$) after 8 h of exposure on 4 July (Figure 4). On 13 September, when high initial concentrations occurred in the water, STX increased by 6.6% (from 442 to 471 ng $L^{-1}$) after 8 h of exposure. In the 0.2 μm filtered water, dissolved STX was reduced by 16.7% on 6 June (from 24.0 to 20.0 ng $L^{-1}$) and 13.2% on 13 September (from 442 to 398 ng $L^{-1}$), while no effects were seen on 4 July. No changes in dissolved, extracellular STX concentrations were observed for samples incubated outdoor in the shade (no direct sunlight).

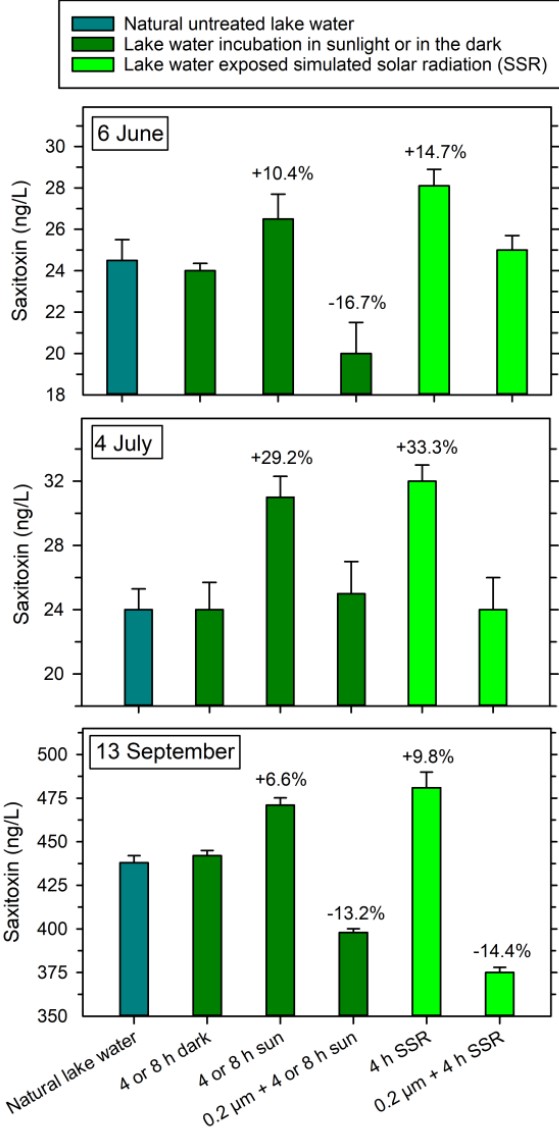

**Figure 4.** Changes in dissolved saxitoxin in natural Lake Arresø water and in 0.2 μm filtered lake water incubated in SaWa® Bags after exposure to natural sunlight, simulated solar radiation (SSR) or incubation in the dark in 2021. On 6 June, the exposure time in sunlight or dark was 4 h, while the exposure time was 8 h on 4 July and 13 September. Incubation in SSR incubator was 4 h at all three dates. The effect of treatments being significantly different from untreated natural lake water are shown as % change (*t*-test, *p* < 0.05%). Quantification was conducted by ELISA technique. Means ± 1 STD are shown (*n* = 3).

Exposure to the simulated solar radiation increased the concentrations of dissolved STX by 14.7% on 6 June (from 24.0 to 28.1 ng L$^{-1}$), 33.3% on 4 July (from 24.0 to 32.1 ng L$^{-1}$) and 9.8% on 13 September (from 442 to 481 ng L$^{-1}$), while a reduction in dissolved STX in the 0.2 μm filtered water only was observed in September (14.4% reduction, from 442 to 375 ng L$^{-1}$) (Figure 4). No effects on dissolved STX were observed in the dark-incubated samples.

The number of detectable *sxtA* gene copies was significantly influenced by both the natural and simulated sunlight exposure. After 4 h of natural or simulated sunlight on 6 June, the copy number was reduced to 5.7% in natural sun (from 626 to 36 copies mL$^{-1}$) and 2.7% (from 626 to 17 copies mL$^{-1}$) in the simulated sunlight (Figure 5). On 4 July, 8 h natural sunlight or 4 h simulated sunlight caused a significant reduction in *sxtA* to 0.33% and 0.16% of the initial number of 24.6 × 10$^3$ copies mL$^{-1}$ to 81 and 41 copies mL$^{-1}$, respectively. An even higher reduction to 0.051% (8 h natural sunlight) and 0.053% (simulated sunlight) was found on 13 September, corresponding to reduction in the original number of 801 × 10$^3$ copies mL$^{-1}$ to 367 and 41 copies mL$^{-1}$, respectively.

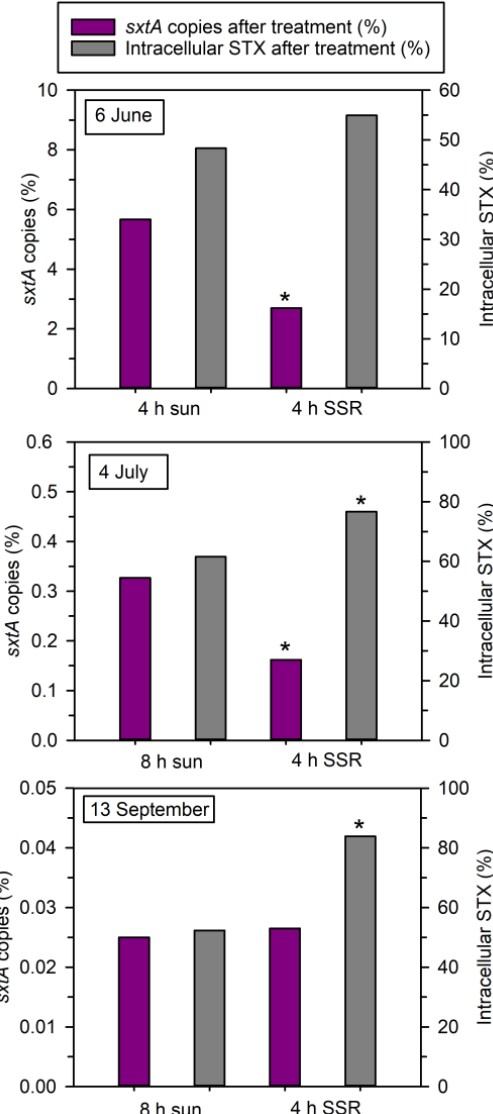

**Figure 5.** Effect of exposure by natural sunlight or simulated solar radiation (SSR) on *sxtA* copy numbers and intracellular STX concentrations in natural lake water. *sxtA* copies (left y axis) and STX concentrations (right y axis) remaining after end of the incubations are shown as % relative to untreated, natural lake water. Exposure time is indicated. Treatments being significantly different from natural lake water are indicated by * (*t*-test, *p* < 0.05%). Quantification was conducted by ELISA technique. Means ± 1 STD are shown (*n* = 3).

The intracellular STX pools were less affected by light as compared to the *sxtA* copies. On the three dates, natural sunlight exposure reduced the intracellular concentration to between 48.3% on 6 June (from 35.5 to 17.1 fg per *sxtA* copy) and 61.6% on 4 July (from 0.47 to 0.29 fg per *sxtA* copy) of the initial concentration (Figure 5). A lower effect was observed for the simulated solar radiation exposure, which reduced the initial pools to 83.7% on 13 September (from 0.63 to 0.53 fg per *sxtA* copy) and 76.7% on 4 July (from 0.47 to 0.36 fg per *sxtA* copy) but a reduction to 54.9% was observed on 6 June (from 35.5 to 19.6 fg per *sxtA* copy).

### 3.5. Microbial Degradation of STX Analogs in the Lake Water

The microbial potential for the degradation of the four STX analogs GTX2/3, NEO, dcNEO and dcSTX was tested on two occasions. As observed for STX, only a partial degradation of these analogs occurred. On 17 August, the highest percentage degradation was found for dcSTX, which was reduced by 72% to 1.54 ng L$^{-1}$ after 75 h (Figure S2). A lower reduction occurred for NEO (51.3% decline to 10 ng L$^{-1}$ at 75 h) and for both GXT2/3 (33.6% decline to 13.1 ng L$^{-1}$) and dcNEO (27.5% decline to 159 ng L$^{-1}$) after 75 h.

Relative to 17 August, on 15 September the initial concentration of NEO was about 10-fold higher, while the dcNEO concentration only was about 25% of that in August (Figure S3). The other two analogs were similar in concentrations on both dates. As in August, a variable reduction in the different STXs was observed, resulting in a reduction ranging from 41.6% (GTX2/3) to 56.3% (dcSTX) after 96 h.

## 4. Discussion

The presence of cyanobacteria in freshwaters can cause water quality concerns and, especially, their production of cyanotoxins may pose a serious risk to human health when present in drinking water reservoirs and in recreational waters [12]. Most studies on the degradation of cyanotoxins have focused on different physicochemical treatments, such as photocatalysis, ozonation and chlorination [15,21]. The microbial degradation of cyanotoxins has also been demonstrated, especially for microcystins, e.g., by He et al. [28]. For STXs, little research on degradation by photochemical approaches has been conducted, as stressed by Kurtz et al. [29], but no knowledge on the microbial capacity for degradation is available. In this study, we provided the first evidence on the microbial potential for the degradation of STX and selected STX analogs.

### 4.1. Concentrations of Dissolved STX and STX Analogs

Changes were observed in the concentrations of dissolved, extracellular STX in Lake Arresø water across the several months and two years, 2020 and 2021, investigated in this study. While most of the dissolved STX concentrations measured were within the range of 16 to 45 ng L$^{-1}$, a high value of 442 ng L$^{-1}$ was observed on 13 September 2021. This indicated that periodically high concentrations may occur, e.g., related to blooms in the lake. The concentrations observed in Lake Arresø were lower than typical STX concentrations measured in lake water in the USA (except for the single high concentration in Lake Arresø) [30]. In the American survey of 1161 lake water samples, STX was detected in 96 samples with concentrations ranging from 70 to 269 ng L$^{-1}$, and this was associated with a high abundance of various cyanobacteria.

Seasonal changes in dissolved, extracellular STX and STX analogs have only been reported in a few studies. In the Brazilian Alagados reservoir, STX and various GTXs together made up concentrations of 5–51 ng L$^{-1}$ in 2007–2008 [13], while concentrations of 0.36–5.3 μg L$^{-1}$ (sum of STX, NEO and various GTXs) were measured in 2013–2014 [14]. The high concentrations were assumed to be caused by blooms in the reservoir by *R. raciborskii*. Comparable concentrations of STX were determined for the Itupararanga reservoir in São Paulo State, Brazil, by Moraes et al. [31] (0.02–0.23 μg L$^{-1}$) and Machado [4] (0.04–0.21 μg L$^{-1}$). The occurrence of different STX analogs as free dissolved compounds in freshwaters were also recently studied in 274 lakes in France, Canada and the UK [32].



STX was always the major compound (concentrations up to 100 ng L$^{-1}$), followed by NEO and dcSTX. In a few cases, STX co-occurred with dcSTX or NEO.

### 4.2. Abundance of STX Producers and Intracellular Pools of STXs

Assuming that each cyanobacterial genome only has one copy of the *sxtA* gene, the observed density of STX-producing cells in Lake Arresø ranged from $0.6 \times 10^3$ copies mL$^{-1}$ (6 June 2021) to $5.6 \times 10^6$ copies mL$^{-1}$ (15 September 2020), which was lower than the previously estimated *sxtA* copy numbers of $49 \times 10^6$ to $117 \times 10^6$ copies mL$^{-1}$ in this lake in late summer in 2018 [8]. The high density of STX producers in the lake most likely reflected the numerous colonies of filamentous cyanobacteria in the lake, as observed by microscopy by Podduturi et al. [8]. In their study, the dominant STX-producers in the lake were identified as the filamentous genera *Dolichospermum*, *Cuspidothrix*, *Phormidium* and *Planktolyngbya*. Supporting that a high number of *sxtA* copies may occur during blooms in freshwater, Al-Tebrineh et al. [33] observed >10$^9$ *sxtA* copies per mL in a eutrophic dam in New South Wales.

The intracellular content of STX per volume of water covaried with the concentration of dissolved, extracellular STX, suggesting that diffusion or release due to cell mortality were controlling the level of dissolved STX in the water, as mentioned for other cyanotoxins by Manganelli [34]. Yet, the cell-specific content of STXs varied significantly (from 0.03 to 35.5 fg per *sxtA* copy) and seemed to be independent of the dissolved, extracellular STX concentration. Previous analysis of the cell-specific STX content in potential STX-producing cyanobacteria in five Danish lakes (including Lake Arresø) indicated a content of 9 to 142 fg STX per cell or colony based upon microscopy [8]. Corresponding values were obtained in a freshwater reservoir, in which 5–28 fg STX were determined per filament (trichome) of *R. raciborskii* [7]. The higher cell-specific values in these two studies, as compared to the present study, might be related to the difference in cell numbers in the colonies vs. single cells or might reflect different rates of STX production. Another reason may be large phenotypic differences, e.g., the size of cells or trichomes between different strains. For example, for *R. raciborskii*, Miotto et al. [35] observed that under laboratory conditions, strain LP2 had a higher quota of STX than strain LP1 (31.03 vs. 18.77 μg L$^{-1}$, respectively). Furthermore, variability in the concentrations of STX can be observed in the same strain. Thus, Ramos et al. [36] noticed a variation in STX concentrations in cultures of the strain LP2, ranging from 2.0 to 15.0 μg L$^{-1}$, when grown at different salinities. According to the authors, variations in salinity introduced stress, but factors influencing the STX synthesis were complex.

Studies of STX regulation and production in marine dinoflagellates have shown that the genetic control of STX biosynthesis is poorly understood and may be controlled by both abiotic (e.g., nutrient stress, temperature and light) and biotic processes (e.g., grazing, allelopathic interactions and symbiotic relationships) [1]. The exposure to natural or simulated sunlight in our study also indicated that solar light had a major impact on the stability of the *sxtA* gene (see below).

### 4.3. Degradation of STX by Lake Water Microorganisms

Despite the fact that previous studies have concluded that STX is not biodegradable by freshwater bacteria [16,37,38] and that physio-chemical procedures are required for the removal of STX from water [15], the present study showed that indigenous microorganisms in Lake Arresø water could reduce dissolved, extracellular SXT levels at ambient concentrations (18 to 52 ng L$^{-1}$) by 40 to 59% after up to 85 h incubation. Thus, although only a partial degradation was observed, the results indicated that microbial populations in the lake were capable of degrading STX. The reduction in the other STX analogs (GTX, NEO, dcNEO and dcSTX) confirmed a partial microbial degradation or transformation (reduction of the various STX analogs from 28 to 72%) by lake water bacteria. Considering the large range in initial concentrations, from about 5 ng dcSTX L$^{-1}$ to 243 ng NEO L$^{-1}$, the change in concentrations of the STX analogs during the incubations appeared not to be

controlled by the amounts of the specific analog. In another study, sand filters (constructed with sand from a freshwater treatment facility and enriched with extracts of *A. circinalis* and containing STX and STX analogs at 0.7–20 µg L$^{-1}$) were shown to partially convert STX to more toxic GTX variants, while concentrations of other STX analogs were reduced [18]. Microbial reduction in the STX analog gonyautoxin (GTX) by freshwater bacteria in subsurface drainage water was examined by Jones and Negri [17] who reported that half of the GTX (obtained from extract of the cyanobacterium *Anabaena circinalis*) was reduced after 9 to 28 days. As in our study, a complete reduction in the STX analog was not observed.

Speculatively, the partial degradation of STX and STX analogs might be related to the nutritional constraints of the bacteria in our tests, but the presence of specific bacterial strains with catabolic potential for the degradation of STXs may also be essential. Donovan et al. [39] tested the degradation of various STXs by marine bacteria isolated from the digestive gland of the blue mussel *Mytilus edulis* during the summer season in Atlantic Canada. When cultivating some of the isolated bacteria in marine broth with extract of the toxin-producing marine dinoflagellate *Alexandrium tamarense*, the toxicity of STX and NEO (determined by a combined mouse toxicity test and HPLC analysis) was reduced after three days by about 95%, while the toxicity of six remaining STXs was only partially reduced. The degradation appeared to be mainly related to the metabolically versatile marine bacterium *Pseudoalteromonas haloplanktis* [40].

The catabolic processes involved in the degradation of STX and its analogs remain to be identified, but they might involve enzymes in the purine or arginine metabolism as suggested by Donovan et al. [40]. In this study, the detection and isolation of STX-degrading lake water bacteria in STX-enriched liquid and agar media was not successful. Speculatively, the added concentration of STX (16 µg L$^{-1}$) could have been too low to induce STX-degrading enzymes in the bacterial cultures, since in the study by Donovan et al. [39], growth of STX-degrading microorganisms was observed when an STX concentration of 22 mg L$^{-1}$ and nutrients were added in the marine broth (peptone and yeast extract).

*4.4. Effects of Natural and Simulated Solar Radiation on STX and Its Biosynthesis*

The exposure to natural and simulated sunlight caused the degradation of dissolved, extracellular STX, the release of intracellular STX from assumed damaged cells, and affected the *sxtA* core gene involved in the biosynthesis of STX and its analogs. Photocatalysis of STX and other cyanotoxins has been applied for the large-scale production of potable water, often with $TiO_2$ as a catalyst [15,21]. Cyanotoxins may also be degraded by both direct solar photolysis (absorption of sunlight) or by sensitized photolysis, in which reactive molecules such as hydroxyl radicals, singlet oxygen, hydrogen peroxide or other reactive species are produced (review by Kurtz et al.) [29]. In lake water, such reactive molecules can be formed by solar exposure of natural dissolved organic matter (DOM). Supporting the importance of sensitized photolysis, a recent study indicated that STX and GTX2/3 appeared to be degraded by sensitized rather than direct photolysis [41]. In the present study, it remains to be examined whether the observed reduction in dissolved, extracellular STX by 13–17% in the filtered water after exposure to natural and simulated sunlight was caused by sensitized or direct photolysis. The eutrophic Lake Arresø is assumed to contain sufficiently high concentrations of DOM for the formation of radical species, but no data on DOM in the lake are available.

In addition to solar radiation, other non-biological processes might have affected the observed changes in the concentrations of STX and STX analogs. Alkaline pH and high temperatures (>90 °C) have been shown to convert some STX analogs, while STX appears rather stable at temperatures <90 °C and at low pH [42]. We assumed that the temperature in our experiments (up to about 40 °C) and the pH of the lake water at the sampling time (pH of 7.4 to 7.8; data not shown) did not affect the concentrations of STX, but they might have modified the concentration of some STX analogs. More studies are needed to confirm this.

The release of intracellular STX of 16–52% leading to an increase in dissolved, extracellular STX of 7–33% by exposure to natural and simulated sunlight in natural lake water

samples indicated that STX-producing cyanobacteria released the STX into the water, most likely due to light-induced cell damage. Light-mediated mortality and severe damage of cell membranes and internal structures were observed in the STX-producing cyanobacterium *R. raciborskii* by Noyma et al. [43] after 6 h exposure to natural subtropical UVA and UVB light intensities. Similarly, when the cyanobacterium *Microcystis aeruginosa* was exposed to alternating, naturally realistic UVB and dark cycles, the toxin microcystin was released [44]. SaWa® Bags have been documented to inactivate different microbial classes such as bacteria (*Escherichia coli*, *Pseudomonas aeruginosa*, and *Vibrio cholerae*), viruses (MS2 bacteriophages) and protozoa (*Cryptosporidium parvum* oocysts), with the mechanism of action being a synergistic effect of UV-induced DNA damage (by UVB), photo-oxidative destruction (by UVA) and thermal inactivation (see www.4lifesolutions.com (accessed on 2 November 2022)). The release of STX in this study thus could be caused by cell damage rather than a deliberate excretion of the toxin, but further studies are needed to conclude whether the irradiance level, increased water temperature or another environmental stressor was the major effecting process.

In addition, the observed reduction in copy numbers of the *sxtA* gene (94.3 to 99.95% reduced amplification in the qPCR assay) after exposure to natural and simulated solar radiation stressed that the expression of this gene was severely affected by the sunlight. UV is known to damage DNA by the production of pyrimidine dimers, also known as photodimers [45], but intense photolysis by UV radiation is assumed to cause a general cell destruction [46]. Cordeiro-Araújo et al. [42] found that the expression of specific *mcy* genes in the biosynthesis of microcystin (another class of toxins produced by freshwater cyanobacteria) was either stimulated or reduced by UVB exposure. Similarly, when Beamud et al. [22] exposed *R. raciborskii* to UVA and UVB radiation, growth and morphology was affected, but expression of the *sxtU* gene and intracellular pools of STX were enhanced. In our study, the variable level of reduction in *sxtA* expression at different sunlight exposures and at different sampling points could speculatively reflect attempts to improve resilience to different types of environmental stress [23].

The observed reduction in *sxtA* copy numbers showed that light exposure compromised the DNA in the cells, but the actual effect of UV dosimetry on the level of DNA damage was not investigated. A previous study by Theitler et al. [47] showed that the level of DNA damage in *E. coli* increased with UV exposure time, i.e., with an increased UV dose. In addition, the DNA damage significantly increased with a combined synergistic effect of heat and UV irradiation, since heat accelerated the DNA damage as compared to UV irradiation only. These findings agree with the results shown in this study. While the temperature conditions were similar under natural and simulated solar radiation, 4 h exposure to simulated solar radiation with a high UV irradiance showed increased or similar DNA damage as compared to 8 h exposure to natural solar irradiance (with a lower UV irradiance; Figure 5). Yet, the exact effect of UV and heat dosimetry on DNA damage in cyanobacteria needs further investigation. Whether DNA repair mechanisms later might have been able to reestablish the *sxtA* gene and other functional genes is uncertain [43].

## 5. Conclusions

Increasing global temperatures in association with the discharge of nutrients into natural freshwaters may lead to more frequent future episodes of cyanobacterial blooms, and such blooms introduce the risk of a stimulated production and release of saxitoxin and other cyanotoxins into the water. So far, no knowledge on the environmental fate of saxitoxin in natural waters has been available, but this first documentation of biological and sunlight-mediated degradation or transformation of the toxin showed that the presence of STX may be influenced by microbial catabolic activity and by different solar irradiance levels. To better predict the occurrence and stability of STX, e.g., in lake water, specific STX-degrading bacteria and their metabolic capacity should be identified. Further, the actual in situ effects of sunlight on the stability of STX and its biosynthesis should be examined.

**Supplementary Materials:** The following supporting information can be downloaded at: https://www.mdpi.com/article/10.3390/w14213556/s1. Figure S1. Density of bacteria in STX degradation experiment on 28 July, 2020. Mean number of bacteria in 10 randomly selected microscope fields ± 1 STD are shown; Figure S2. Microbial degradation of four different STX analogs determined by HPLC quantification. Lake water was collected on on 17 August, 2020. Blue columns represent initial concentration in the lake water. Means ± 1 STD are shown (*n* = 3); Figure S3. Microbial degradation of four different STX analogs determined by HPLC quantification. Lake water was collected on 15 September, 2020. Blue columns represent initial concentration in the lake water. Means ± 1 STD are shown (*n* = 3).

**Author Contributions:** Conceptualization, N.O.G.J. and C.F.M.; methodology, N.O.G.J., R.P., C.F.M., T.J. and M.d.A.B.M.; writing—original draft preparation, N.O.G.J.; writing—review and editing, N.O.G.J., R.P., C.F.M., T.J. and M.d.A.B.M. All authors have read and agreed to the published version of the manuscript.

**Funding:** Danish Agency for Science and Higher Education, Ministry of Higher Education and Science, Denmark, Grant 9096-00040B.

**Institutional Review Board Statement:** Not applicable.

**Informed Consent Statement:** Not applicable.

**Data Availability Statement:** Not applicable.

**Acknowledgments:** Susanne Iversen is acknowledged for skillful assistance with the molecular work.

**Conflicts of Interest:** The authors declare no conflict of interest.

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
