# Peer review of "Fate of Saxitoxins in Lake Water: Preliminary Testing of Degradation by Microbes and Sunlight"

_water, doi:10.3390/w14213556_

Round 1
Reviewer 1 Report
MS number: 1971669
The Review of the Paper: Fate of saxitoxins in lake water: Preliminary testing of degrada-2 tion by microbes and sunlight
Authors: Niels O. G. Jørgensen1, Raju Podduturi1, Charlotte F. Michelsen, Thea Jepsen, Munique de Almeida Bispo Moraes
Paper deals with the very interesting, modern, and innovative topic on the saxitoxins in lake water in preliminary testing of degrada-2 tion by microbes and sunlight. The title clearly reflects the content of the paper. In my opinion, data quality is high, the interpretation of data – is very good, and the importance of work- high. Methods are suitable for the study, all text is written in a clear way but requires some minor corrections that should be incorporated into the manuscript before publication. The paper deal with an interesting topic and can be published in WATER after careful revision. Detailed comments, specific issues, and some suggestions are presented below and may help the authors in their revision.
Abstract and Introduction and Material and Methods: these parts of the paper were described in detail, especially the methodology, and I have only minor comments on the Introduction part of the manuscript.
· Introduction, lines 59-60: the size of fonts should be changed
· Introduction, line 65: R. Raciborskii - since the species name is used here for the first time, it should be written as a full species name
Figures: are clear and show the trend in the experiment results in a proper way.
Discussion: was divided into the same parts as the Results section, which allowed to organize the content of these parts. I have no comments on the Discussion except the citations (suggestion below)
Conclusions: Because the study is preliminary and was carried out only in one lake, it should be emphasized that more studies in other lakes and maybe different types of water bodies are needed to show/ confirm the same results that water bacterioplankton and sunlight can modify STX by degra-30 dation and cell destruction, and that biosynthesis of STX may be inhibited by exposure to sunlight.
Literature: In my opinion, more recent papers (if possible) should be cited in the introduction and also in the discussion. If not, please provide an explanation in the Discussion or Introduction on the limitation of recent literature on the topic.
Author Response
We appreciate the positive comments on the manuscript. The following amendments have been done (line numbers refer to the original manuscript):
A total of 8 recent literature references have now been added in the Introduction and Discussion. Unfortunately, degradation of saxitoxin and its analogs has rarely been studied, and therefore we cannot find references that directly substantiate our own results.
Lines 59-60: Font size has been corrected.
Lines 65: Name of R. raciborskii has been written in full species name (Raphidiopsis raciborskii).
Regarding the comment on more studies in other lakes and maybe different types of water bodies: In the revised manuscript, we have added information on the possible role of water pH and temperature on stability of SXT and STX analogs. Further, we have added information on a possible synergetic effect from a combined solar and temperature degradation of STX.
Since there are no other published studies on biological STX degradation or on in situ solar degradation in lake water, we are reluctant to speculate or hypothesize on effects of other cyano-toxins in other types of lakes. We think this must await more experimental studies before conclusions can be made on such effects.
Reviewer 2 Report
Authors have made several interesting research experiments regarding STX concentration and degradation (microbial and UV light). Overall significance of the findings will be of interest to the readers and could contribute to the general knowledge on the subject. I imagine that it must have been difficult to carefully explain the research - generally it seems a bit chaotic, varying in the number of samples, which tests were done with which samples, how detailed was the research at some points etc. It takes time to understand the whole process, and if authors have another way of making the research and what they did more clearly, it would be advised. Perhaps a table summarizing the analyses per dates that precedes the methodology section would be a good idea for an overview. Perhaps, similarly it can be done for results as well, summarizing the findings in a table.
Also, please find attached additional few comments of the manuscript.
Given that the paper raises intriguing questions based on the results provided, I recommend that the paper is published in your journal after some minor revisions.

Author Response
We appreciate the positive comments on the manuscript by Reviewer 2. The following amendment have been done (line numbers refer to the original manuscript):
We realize that an overview of analyses etc. might help readers better follow the work being done. Now, a table (Table 1) summarizing sampling, analyses and treatments has been included at the end of Material and Methods.
Regarding the comment on more studies in other lakes and maybe different types of water bodies: In the revised manuscript, we have added information on the possible role of water pH and temperature on stability of SXT and STX analogs. Further, we have added information on a possible synergetic effect from a combined solar and temperature degradation of STX.
Since there are no other published studies on biological STX degradation or on in situ solar degradation in lake water, we are reluctant to speculate or hypothesize on effects of other cyano-toxins in other types of lakes. We think this must await more experimental studies before conclusions can be made on such effects.
Reviewer 3 Report
This study presents data on the presence of saxitoxins and their derivatives in the water of a Danish eutrophic lake, and preliminarily investigates the potential for degradation or increase of dissolved and intracellular saxitoxins by lake bacteria, as well as by treatments with natural and artificial light. Additionally, it points out the potential of light treatments to damage the sxtA gene, involved in the synthesis of saxitoxin.
The main results suggest that: lake bacteria can reduce up to 59% of dissolved saxitoxin in a period of 3 days; sunlight can reduce up to 45% of intracellular saxitoxin and increase dissolved saxitone by up to 33%; and that light can damage up to 95% of the sxtA gene.
The study was conducted with a creative and adequate methodological approach to generate preliminary data on saxitoxin degradation mainly in an ecological context, but also in an applied context, as possible procedures for saxitoxin reduction in water, once more complementary studies are conducted. The results are very rich for generating hypotheses for further studies on this topic, being therefore quite relevant.
Some specific comments and questions (highlighted in the revised version):
1) Did you do a control treatment, with only the sterile filtered sample, without inoculation with the sample filtered by a 1.3 um filter? If so, please explain, show and comment. If not, justify. Could the concentration of saxitoxin in this experiment, carried out in the dark, be reduced by any process other than bacterial degradation and light?
2) Throughout the results you repeat information from the methodology (highlighted in yellow in the revised version). Consider simplifying or removing this information.
3) About variation of STX in R. raciborskii (line 450): I would say that it may also be related to the large phenotypic differences (eg size of trichomes/cells) between different strains of R. raciborskii (see Brazilian studies: MIOTTO et al. In HYDROBIOLOGIA, 802 (2017) and RAMOS et al. In HARMFUL ALGAE, 103 (2021)).
4) See also little observations highlited in the text.
5) Did you find any data relating the amount of light (photon flux density), including UV, and the level of DNA damage? Did your experiments generate results in this regard?
6) Final comments:
The issue of cyanotoxin degradation is very interesting and has the potential to support new hypotheses and studies, such as: verifying relationships between bacterial density and diversity and the efficiency of degradation OR on eventual processes in cometabolism, using different culture media, OR STILL, if physicochemical variables of the water can influence this efficiency. The format of your manuscript allows for more speculative and hypothetical questions like these to be included in the discussion and conclusion. You have already done some, but I strongly suggest that you add others, as the results suggest so.

Author Response
We acknowledge the constructive comments to the manuscript by reviewer 3. Below, please find our replies to the comments.
1) Did you do a control treatment, with only the sterile filtered sample, without inoculation with the sample filtered by a 1.3 um filter? If so, please explain, show and comment. If not, justify. Could the concentration of saxitoxin in this experiment, carried out in the dark, be reduced by any process other than bacterial degradation and light?
Reply: In our study, concentrations of STX in lake water samples without microorganisms (0.2 µm filtered) remained unchanged for at least 6 days at room temperature in the dark. Further, we realize that non-biological processes such as alkaline pH and high temperatures can convert some STX analogs (>35°C), while STX appears rather stable, and a reference has been added on this in the Discussion. Since pH of the present lake water was non-alkaline (pH of 7.4 to 7.8) and temperatures were 20-22°C during the biological experiments, we assume that no chemical conversion of STX occurred in our study. In the solar exposure experiments, the temperature reached 40°C - 42°C, but we cannot document whether this affected the chemical stability of STX.
The information on stability of STX has been added in Results (lines 318-322 in the revised MS) and in Discussion (lines 530-536 and lines 567-579 in the revised MS). A literature reference has also been added.
As for effect of temperature, additional information has been given in M&M, lines 247-248 in the revised manuscript
2) Throughout the results you repeat information from the methodology (highlighted in yellow in the revised version). Consider simplifying or removing this information.
Reply: The high-lighted texts in the pdf file have been revised or removed. However, in the abstract, we have kept the last sentence, since it summarizes the main results of the study. Many researchers like such a concluding sentence. In section 3.1 in Results, yellow text in two sentences is kept since this is new information (sxtA gene is involved in synthesis of both STX and STX analogs) and to explain why sxtA numbers were related to all studied STX analogs.
3) About variation of STX in R. raciborskii (line 450): I would say that it may also be related to the large phenotypic differences (eg size of trichomes/cells) between different strains of R. raciborskii (see Brazilian studies: MIOTTO et al. In HYDROBIOLOGIA, 802 (2017) and RAMOS et al. In HARMFUL ALGAE, 103 (2021)).
Reply: We acknowledge this comment and have added the following text in the manuscript (lines 459-467 in the revised MS):
Another reason may be large phenotypic differences, e.g., size of cells or trichomes, between different strains. For example, for R. raciborskii, Miotto et al. [35] observed that under laboratory conditions, strain LP2 had a higher quota of STX than strain LP1 (31.03 vs. 18.77 μg L-1, respectively). Furthermore, variability in the concentrations of STX can be observed in the same strain. Thus, Ramos et al. [36] noticed a variation in STX concentrations in cultures of the strain LP2, ranging from 2.0 to 15.0 μg L-1, when grown at different salinities. According to the authors, variations in salinity introduce stress, but factors influencing the STX synthesis are complex.
4) See also little observations highlited in the text.
Reply: Text has been revised or commented.
5) Did you find any data relating the amount of light (photon flux density), including UV, and the level of DNA damage? Did your experiments generate results in this regard?
Reply: We have added the following text in lines 567-579 in the revised manuscript:
The observed reduction in sxtA copy numbers shows that light exposure compromised DNA in the cells, but the actual effect of UV dosimetry on the level of DNA damage was not investigated. A previous study by Theitler et al. [48] showed that the level of DNA damage in E. coli increased by UV exposure time, i.e., by increased UV dose. In addition, the DNA damage significantly increased by a combined synergistic effect of heat and UV irradiation, since heat accelerated the DNA damage as compared to UV irradiation only. These findings agree with the results shown in this study. While the temperature conditions were similar under natural and simulated solar radiation, 4 h exposure to simulated solar radiation with a high UV irradiance showed increased or similar DNA damage as compared to 8 h exposure to natural solar irradiance (with a lower UV irradiance; Figure 5). Yet, the exact effect of UV and heat dosimetry on DNA damage in cyanobacteria needs further investigation. Whether DNA repair mechanisms later might have been able to reestablish the sxtA gene and other functional genes is uncertain [44].
6) Final comments:
The issue of cyanotoxin degradation is very interesting and has the potential to support new hypotheses and studies, such as: verifying relationships between bacterial density and diversity and the efficiency of degradation OR on eventual processes in cometabolism, using different culture media, OR STILL, if physicochemical variables of the water can influence this efficiency. The format of your manuscript allows for more speculative and hypothetical questions like these to be included in the discussion and conclusion. You have already done some, but I strongly suggest that you add others, as the results suggest so.
Reply:
As mentioned above, in the manuscript we have included discussion on whether other environmental factors, such as alkaline pH and temperature, which have been shown to change or convert some STX analogs in laboratory experiments, might have affected our observed changes in concentration. However, since the literature on this topic is very limited, we do think we should add unsubstantiated hypotheses here, but await more studies in the area to be published.